# GermVarX: A Robust Workflow for Joint Germline Variant Exploration in whole-exome sequencing cohorts

**Thao Thi Phuong Nguyen**[1]*, **Dung Duc Nguyen**[1], **Thuy Van Mai**[2], **Dung Khoi Nguyen**[3], **Tung Dang Nguyen**[4], **Ngoc Thi Minh Truong**[1], **Hanh Hong Ha**[5], **Trang Thi Ha Tran**[6]

**1** Institute of Information Technology, Vietnam Academy of Science and Technology, Hanoi, Vietnam, **2** Hanoi University of Public Health, Hanoi, Vietnam, **3** Electric Power University, Hanoi, Vietnam, **4** Post and Telecommunications Institute of Technology, Hanoi, Vietnam, **5** Institute of Biology, Vietnam Academy of Science and Technology, Hanoi, Vietnam, **6** VinUni Bigdata Research Institute, VinUniversity, Hanoi, Vietnam

* thaontp@ioit.ac.vn

## Abstract

Accurate identification of germline variants from whole-exome sequencing (WES) data is foundational to population genetics, disease association studies, and clinical genomics. However, variant calling across cohorts poses challenges in scalability, consistency, and reproducibility. We present GermVarX, a fully automated, modular workflow for joint germline variant discovery and exploration in WES cohort studies. A key feature of GermVarX is its implementation of joint variant calling, enabling simultaneous genotyping of multiple samples to produce a single, high-confidence multi-sample VCF, optimized for downstream analyses. Developed with Nextflow DSL2, GermVarX ensures reproducibility, portability, and efficient parallelization across diverse computing environments, including workstations, HPC clusters, and cloud platforms. The workflow integrates two state-of-the-art variant callers—GATK HaplotypeCaller and DeepVariant—with joint genotyping performed via GATK or GLnexus. To increase reliability, GermVarX supports consensus generation between callers, coupled with sample- and cohort-level quality control, functional annotation using the Variant Effect Predictor (VEP), and unified reporting through MultiQC. In addition, it provides PLINK-compatible outputs, facilitating seamless integration with statistical and association analyses. GermVarX delivers a scalable, reproducible, and comprehensive solution for germline variant analysis in large WES studies, supporting consistent and interpretable results for both research and clinical genomics. The source code and usage instructions are available at https://github.com/thaontp711/GermVarX.

**Data availability statement:** The Ashkenazim Trio whole-exome sequencing (WES) data used for accuracy evaluation are publicly available from the NCBI Sequencing Read Archive (https://www.ncbi.nlm.nih.gov/sra/) under the accession numbers SRR2962669, SRR2962692, and SRR2962694. Corresponding Genome in a Bottle (GIAB) benchmark datasets were obtained from the GIAB FTP repository (https://ftp-trace.ncbi.nlm.nih.gov/giab/ftp/release/). The 120 WES samples used for performance assessment are part of an ethics-approved dataset from the Institute of Biology (formerly the Institute of Genome Research). The study was conducted in accordance with the Declaration of Helsinki and approved by the Ethics Committee of the Institute of Genome Research (Approval No. 01-2021/NCHG-HDDD; 26 October 2021). Due to ethical restrictions regarding participant privacy, these data cannot be shared publicly. Data are available from the Ethics Committee of the Institute of Biology (contact via hhhanh@ib.ac.vn) for researchers who meet the criteria for access to confidential data.

**Funding:** This work was supported by the Vietnam Ministry of Science and Technology (grant no. KC4.0-37/19-25) awarded to Nguyen Thi Phuong Thao, Nguyen Duc Dung, Ha Hong Hanh, Nguyen Khoi Dung, and Nguyen Dang Tung, and by the Institute of Information Technology, Vietnam Academy of Science and Technology (grant no. CSCL02.06/24-25) awarded to Nguyen Thi Phuong Thao, Mai Van Thuy, and Truong Thi Minh Ngoc. The funders had no role in study design, data collection and analysis, decision to publish, or preparation of the manuscript. Funder websites: Vietnam Ministry of Science and Technology: https://mst.gov.vn/ Institute of Information Technology, Vietnam Academy of Science and Technology: https://vast.gov.vn/.

**Competing interests:** The authors have declared that no competing interests exist.

## Introduction

The increasing accessibility of Next-Generation Sequencing (NGS) technologies, particularly Whole-Exome Sequencing (WES), has profoundly advanced genetic and genomic research. By targeting protein-coding regions—which harbor the majority of disease-associated variants—WES offers a cost-effective strategy for identifying genetic alterations relevant to inherited diseases, cancer susceptibility, and population-level variation.

For population-scale genetic studies and clinical diagnostics, accurate detection of germline variants across cohorts is critical. This necessitates robust, scalable, and reproducible bioinformatics workflows capable of handling high-throughput data while preserving data integrity, delivering comprehensive variant call sets, and supporting downstream analyses. This need is compounded by the fact that many publicly available datasets—such as case–control studies—are provided only as raw FASTQ files, requiring substantial preprocessing before meaningful interpretation is possible. However, most existing workflows are designed for per-sample analysis, limiting their applicability to cohort-level studies. Examples include Sarek [1], a portable Nextflow-based pipeline; DNAscan2 [2] and hDNApipe [3], which offer both command-line and graphical interfaces; and Galaxy [4], a web-based platform. Only a few pipelines, such as MagicPipeline [5] and star_protocols_wes [6], support joint-calling workflows, but they are predominantly script-based and command-line dependent, lacking key features such as automated orchestration, modular parallelization, full containerization, and scalability across diverse compute environments.

In this study, we present GermVarX, a fully automated workflow for germline variant discovery and exploration in WES cohort studies. A key feature of GermVarX is its use of joint genotyping, where all samples in a cohort are genotyped together rather than individually. This approach improves detection and genotyping of low-frequency alleles by using shared evidence across samples, reducing noise and increasing statistical power. Prior work with GLnexus has shown that in large cohorts, joint calling rescues rare alleles that single-sample or small-batch calling often miss [7]. Similarly, GATK's Best Practices recommend the GVCF + joint genotyping workflow for germline short variants precisely for these benefits [8].

To ensure high reliability, GermVarX integrates two state-of-the-art variant callers: GATK HaplotypeCaller [9], widely regarded as the gold standard for germline variant discovery, and DeepVariant [10], a deep learning-based tool known for its superior accuracy and robustness [11]. Joint genotyping is performed using either GATK or Glnexus [12], and GermVarX further supports consensus variant calling between the two callers to produce a single, high-confidence multi-sample VCF file for the entire cohort. The workflow also includes comprehensive quality control at both sample and cohort levels, functional annotation using the Variant Effect Predictor (VEP) [13], and consolidated reporting with MultiQC [14]. Additionally, PLINK-compatible outputs are provided to facilitate seamless integration with downstream statistical and association analyses [15].

Built using Nextflow DSL2 [16,17], GermVarX supports modular, parallelized execution across all stages of analysis—from raw FASTQ preprocessing to joint variant

calling, annotation and cohort-level quality control. The workflow is designed with an emphasis on reproducibility, portability, and scalability, enabling deployment on local workstations, high-performance computing (HPC) clusters, and cloud-based platforms. Through this design, GermVarX offers a practical, ready-to-use solution that streamlines WES cohort processing and eliminates the substantial effort typically required to develop, validate, and maintain comparable in-house pipelines.

We validated GermVarX on two datasets: (i) a 3-sample WES benchmark dataset with known variant ground truth, and (ii) a 120-sample WES cohort, demonstrating the workflow's reliability, accuracy, and suitability for both small-scale and moderate-scale population studies.

## Materials and methods

GermVarX is an automated and modular workflow designed for germline variant discovery and exploration in WES cohort studies. Implemented in Nextflow DSL2, it supports fully automated execution, a modular architecture, and parallelized task execution across diverse computing environments, including local workstations, HPC clusters, and cloud platforms. Inspired by the design principles of Kuura [18], which demonstrated the benefits of a simple, user-friendly workflow implemented in Nextflow with Docker, GermVarX adopts a containerized approach using Docker to ensure portability, reproducibility, and ease of deployment. The GermVarX workflow comprises two primary phases: (i) per-sample processing and (ii) cohort-level processing and annotation. A schematic overview of the pipeline is shown in Fig 1, and the software tools with versions used in this study are summarized in Table 1.

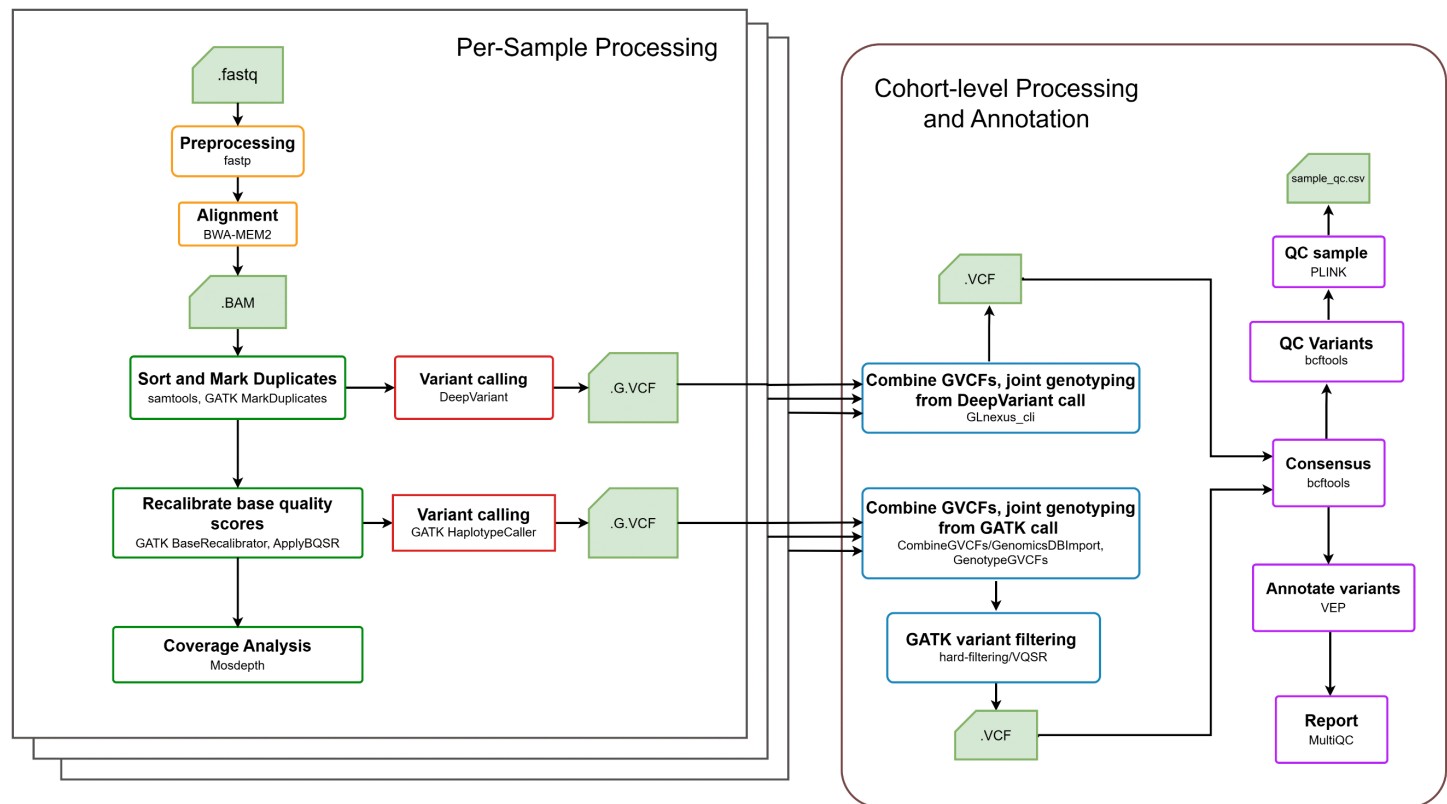

**Fig 1. GermVarX workflow.** The GermVarX workflow comprises two primary phases: per-sample processing and cohort-level processing and annotation.

**Table 1. Software requirements and availability.**

| Software | Version | Availability |
|----------|---------|--------------|
| **Required software** | | |
| Nextflow | 24.10.6 | https://www.nextflow.io/index.html |
| Docker | 28.1.1 | https://www.docker.com/ |
| Python3 | 3.10.12 | https://www.python.org/downloads |
| **Included analysis software** | | |
| FASTQC | 0.11.9 | https://www.bioinformatics.babraham.ac.uk/projects/fastqc/ |
| BWA-MEM2 | 2.2.1 | https://github.com/bwa-mem2/bwa-mem2 |
| GATK4 | 4.2.6.1 | https://gatk.broadinstitute.org/hc/en-us |
| DeepVariant | 1.6.1 | https://github.com/google/deepvariant |
| GLnexus | 1.4.1 | https://github.com/dnanexus-rnd/GLnexus |
| VEP | 114.1 | https://asia.ensembl.org/info/docs/tools/vep/index.html |
| PLINK | 1.9 | https://www.cog-genomics.org/plink |
| Mosdepth | 0.2.4 | https://github.com/brentp/mosdepth |
| SAMTOOLS | 1.15.1 | https://www.htslib.org/ |
| BCFTOOLS | 1.15.1 | https://samtools.github.io/bcftools/howtos/index.html |
| HTSLIB | 1.15.1 | https://github.com/samtools/htslib |
| BEDTOOLS | 2.30.0 | https://bedtools.readthedocs.io/en/latest/ |
| fastp | 0.23.1 | https://github.com/OpenGene/fastp |
| MultiQC | 1.30 | https://github.com/MultiQC/MultiQC |

## Per-sample processing

The initial phase prepares individual samples for joint genotyping by performing quality control, alignment, recalibration, and per-sample variant calling.

- **Input Data**: The workflow accepts paired-end FASTQ files as the primary input.

- **Initial Quality Control**: Raw reads are assessed using FastQC to detect potential issues such as adapter contamination, low-quality bases, or sequence composition biases.

- **Adapter and Quality Trimming**: fastp is used to remove adapter sequences and low-quality bases, improving alignment accuracy and reducing false positives.

- **Read Alignment**: The cleaned paired-end reads are aligned to the reference genome (GRCh38) using BWA-MEM2, producing SAM files, which are converted to BAM format for downstream processing.

- **Sorting and Duplicate Marking**: BAM files were first coordinate-sorted using samtools to enable efficient genomic data access. Subsequently, Polymerase Chain Reaction (PCR) duplicates, which may artificially inflate allele counts, are identified and marked using gatk MarkDuplicates.

- **Base Quality Score Recalibration (BQSR)**: To correct systematic errors in base quality scores, we apply GATK's BaseRecalibrator followed by ApplyBQSR, thereby improving the accuracy of downstream variant calling with the GATK HaplotypeCaller.

- **Per-Sample Variant Calling**: GermVarX employs a dual variant calling strategy for robustness:

  - **GATK HaplotypeCaller:** Generates GVCF files (`.g.vcf`) for each sample, capturing variant and non-variant sites for joint genotyping.

  - **DeepVariant:** Processes BAM files to produce GVCF files using a deep learning-based approach.

 

## Cohort-level processing and annotation

The second phase of GermVarX aggregates per-sample variant data to perform joint genotyping, consensus building, variant filtering, and functional annotation for the entire cohort.

- **Joint Genotyping:** GermVarX supports two joint genotyping strategies based on the variant caller used:

  - **For GATK-based calling:**

    - `CombineGVCFs/GenomicsDBImport`: Supports merging per-sample GVCFs into a combined cohort GVCF. GermVarX uses `CombineGVCFs` as the default approach and provides optional support for `GenomicsDBImport`, which offers improved scalability and efficiency for large cohorts [19].

    - `GenotypeGVCFs`: Performs joint genotyping to generate a multi-sample VCF.

    - **Variant filtering**: GermVarX provides two user-selectable filtering methods: (1) GATK hard filtering, applying best-practice threshold-based filters for SNPs and indels; (2) Variant Quality Score Recalibration (VQSR), a machine learning–based approach that models variant quality and applies probabilistic filtering for SNPs and indels following GATK best practices.

  - **For DeepVariant-based calling:**

    - GLnexus: Performs optimized joint genotyping on DeepVariant GVCFs, as GATK `CombineGVCFs/GenomicsDBImport` is incompatible with DeepVariant outputs.

- **Variant Annotation:** Functional annotation is performed using `VEP` (Variant Effect Predictor) with multiple plugins, including `dbNSFP, CLINVAR`, and `CADD`, providing pathogenicity predictions and gene impact assessment.

- **Consensus Building:** To improve reliability, GermVarX supports consensus callset generation by intersecting variant calls from GATK and DeepVariant. Overlapping variants are retained when they share both genomic position and alleles, producing a high-confidence multi-sample VCF.

- **Variant Quality Control:** Variants from the consensus file are subjected to stringent quality control procedures prior to downstream analyses. Specifically: (i) Genotypes with sequencing depth (DP) < 10, genotype quality (GQ) < 20, or heterozygous calls with allele balance (AB) outside the range of 0.2–0.8 are marked as missing; (ii) Multiallelic sites are excluded; (iii) Variants with a quality score (QUAL) < 20 or a call rate < 90% across samples are removed. These values are used as defaults; however, all filtering thresholds can be easily modified by users (see Table 2 of S1 File).

  At the sample level, summary statistics are computed to assess overall data quality. These include the transition/transversion (Ti/Tv) ratio, heterozygous/homozygous (Het/Hom) ratio, and insertion/deletion (Indel) ratio, all generated using `bcftools stats`. Additional checks for relatedness and sex concordance are performed using PLINK. All quality control results are consolidated into the file `sample_qc.csv`, enabling straightforward filtering and selection of high-quality samples for subsequent analyses.

- **Reporting:** Throughout the workflow, quality metrics are collected from all stages. At completion, `MultiQC` aggregates results from `FastQC, BWA, GATK, VEP`, and `mosdepth` into a single interactive report.

## Portability and reproducibility

GermVarX is implemented in Nextflow DSL2, leveraging features such as process parallelization, container integration, checkpointing, and error handling to ensure efficiency, reproducibility, and fault tolerance. Each tool is encapsulated within a modular Nextflow process, enabling easy updates and configuration across computing environments. To guarantee

reproducibility and minimize installation overhead, GermVarX is fully containerized with Docker, packaging all dependencies into pre-built images.

The workflow employs explicit version control for both pipeline code and software containers, enabling analyses to be reproduced consistently across runs and environments. In addition, GermVarX generates standardized outputs—including VCF, PLINK, and MultiQC reports—providing a stable and transparent basis for downstream analyses in WES cohort studies.

### Setting up and running GermVarX pipeline

GermVarX is implemented as a Nextflow DSL2 workflow and can be executed on any POSIX-compliant computing system. All tools are fully containerized using Docker, thereby ensuring reproducibility, portability, and the elimination of software dependency conflicts. The pipeline requires only minimal setup, consisting of the installation of Nextflow (≥24) and Docker on the target environment.

The general procedure for setup and execution involves:

1. Install and configure Nextflow and Docker on the target system.

2. Clone the GermVarX source code from the GitHub repository.

3. Pull the required pre-built container images and build the custom GermVarX Docker image.

4. Configure input parameters and execution settings in the configuration files provided within the `configuration`/ directory.

5. Run the pipeline using Nextflow with Docker support enabled. A typical command is:

```
nextflow run src/main.nf -profile docker <INPUT> [OPTIONS]
```

By default, GermVarX executes the full pipeline, beginning with paired-end FASTQ files and proceeding through alignment, variant calling, filtering, annotation, and reporting. However, the workflow also supports alternative entry points, accepting BAM, GVCF, or VCF files as input. In such cases, execution commences automatically at the appropriate downstream stage, thereby bypassing earlier processes. Similarly, users can specify the desired final output format (e.g., BAM, GVCF), allowing the pipeline to terminate at a chosen stage.

The protocol described in this peer-reviewed article is published on protocols.io (dx.doi.org/10.17504/protocols.io.3byl48kr8vo5/v1) and is included for printing purposes as S1 File.

### Data acquisition

To evaluate the performance and accuracy of the GermVarX protocol, we utilized two distinct datasets:

- **Ashkenazim Trio (Benchmark Dataset):** Three WES samples (SRA accession IDs: SRR2962669, SRR2962692, and SRR2962694) sequenced on the Illumina HiSeq 2500 platform with approximately 100× coverage using the Agilent SureSelect Human All Exon V5 capture kit were selected. Raw sequencing data are publicly available from the NCBI Sequencing Read Archive (https://www.ncbi.nlm.nih.gov/sra/). Corresponding Genome in a Bottle (GIAB) benchmark datasets (v4.2.1) were obtained from the GIAB FTP repository (https://ftp-trace.ncbi.nlm.nih.gov/giab/ftp/release/).

- **120 WES samples (Computational Performance Evaluation):** To assess the workflow's scalability and performance in a cohort setting, we used 120 WES samples provided by the Institute of Biology (formerly the Institute of Genome Research). The study involving these samples was conducted in accordance with the Declaration of Helsinki and approved by the Ethics Committee of the Institute of Genome Research (Approval No. 01–2021/NCHG-HDDD; 26 October 2021). These data were used exclusively to evaluate the workflow's computational efficiency and resource management.

## Results

### Variant calling accuracy

To assess the biological accuracy of GermVarX, we benchmarked its variant calling performance using three WES samples from the Ashkenazim Trio. We compared three variant calling strategies: (i) GATK join genotyping with hard filtering (GATK-Join), (ii) DeepVariant single-sample calling with GLnexus merging (DV-GLN), and (iii) Consensus, which retains only variants detected by both callers. Because GermVarX performs joint variant calling across samples, benchmarking was conducted against the Genome in a Bottle (GIAB) v4.2.1 truth sets, restricted to high-confidence regions intersecting with the capture targets and shared across all samples.

Variant evaluation was performed using the hap.py toolkit [20], which reports three standard performance metrics based on the Global Alliance for Genomics and Health (GA4GH) benchmarking guidelines for variant calling: *precision*, *recall*, and *F1-score*. These metrics compare the variant calls against a ground-truth benchmark set, where TP (true positives) denotes correctly identified variants, FP (false positives) denotes variants called by the caller but absent from the truth set, and FN (false negatives) denotes true variants that were missed by the caller.

Precision is defined as

$$\text{Precision} = \frac{TP}{TP + FP},$$

recall as

$$\text{Recall} = \frac{TP}{TP + FN},$$

and the F1-score as the harmonic mean of precision and recall:

$$F1 = 2 \times \frac{\text{Precision} \times \text{Recall}}{\text{Precision} + \text{Recall}}.$$

These metrics were computed separately for single-nucleotide polymorphisms (SNPs) and insertions/deletions (INDELs). A summary of results is shown in Fig. 2, with per-sample details provided in S1 Table.

For SNPs, all strategies achieved high accuracy. DV-GLN achieved the highest F1-score (0.9649), balancing exceptional precision (0.9994) with strong recall (0.9327). GATK-Join performed similarly (F1 = 0.9639), with slightly reduced precision (0.9960) but comparable recall (0.9338). The Consensus call set yielded the highest precision (0.9995) and nearly identical recall (0.9326), resulting in an F1-score equal to DV-GLN (0.9649).

For INDELs, performance differences were more pronounced. DV-GLN outperformed other strategies in terms of F1-score (0.9372), driven by its very high precision (0.9927), although its recall (0.8876) was lower than GATK-Join (0.9121). GATK-Join provided balanced performance (precision = 0.9058, recall = 0.9121, F1 = 0.9090). The Consensus call set maintained the highest precision (0.9983) but at the cost of recall (0.8635), yielding an intermediate F1-score (0.9260).

### Computational performance

To assess computational requirements, GermVarX was benchmarked on WES datasets ranging from dozens to hundreds of samples. As a representative test, we executed the full pipeline on 120 samples (403 GB of raw data, 50× coverage) using a system with 56 CPU cores, 128 GB of RAM, and Ubuntu 24.04 LTS. The run completed in 77.8 hours of wall-clock time and required approximately 3.5 TB of storage across input, work, and output directories.

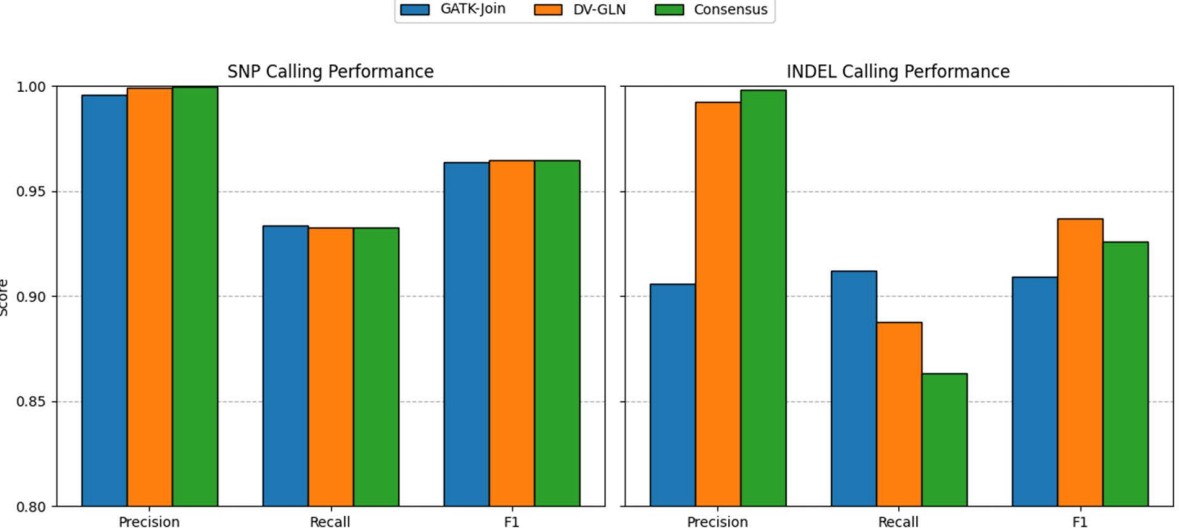

**Fig 2. GermVarX variant benchmarking.** Benchmarking of GATK-Join, DV-GLN, and Consensus strategies for SNPs and INDELs. DV-GLN and Consensus show highest SNP accuracy, while GATK-Join provides balanced INDEL performance.

Most of the persistent storage was consumed by preprocessing (FASTQ trimming and BAM file generation). Upon completion, up to 2.5 TB of temporary files in the work directory can be safely removed, unless re-runs from intermediate states are needed.

The cumulative CPU and wall-clock hours, calculated as the sum of all task runtimes across the 120-sample dataset, are summarized in Table 2. It is important to note that these per-stage totals represent aggregate task time rather than actual elapsed time, which for the complete GermVarX execution was 77.8 hours. Among the stages, preprocessing of FASTQ and BAM files—including trimming, read alignment, and duplicate marking—was the most resource-intensive step (585.8 CPU hours, 226.3 wall-clock hours). This was followed by GATK-based variant calling (159.1 CPU hours, 93.1 wall-clock hours) and DeepVariant-based variant calling steps (186.1 CPU hours, 51.1 wall-clock hours). Other stages—including quality control, annotation, and consensus building—were comparatively lightweight. Peak memory usage remained modest, not exceeding 21 GB across all processes, demonstrating that GermVarX can be efficiently deployed on mid-range HPC clusters or cloud-based environments.

## Discussion

Unlike many existing pipelines that primarily support independent per-sample variant calling, GermVarX provides an automated, end-to-end join genotyping workflow that produces a unified multi-sample VCF – an essential requirement for genetic association studies, case–control designs, and population-scale genomics.

The key differences between GermVarX and existing WES workflows are summaried in Table 3. The table includes Sarek as a representative per-sample workflow, while MagicPipeline and star_protocols_wes are two available protocol for WES cohort studies.

Our benchmarking demonstrates that GermVarX delivers highly accurate variant calls across multiple strategies, with particularly strong performance for SNPs. DeepVariant with GLnexus (DV-GLN) achieved the best F1-scores but had lower recall in 3-sample GIAB WES dataset, consistent with reports that GLnexus performs best in large cohorts where allele frequency information improves sensitivity [12]. Similarly, the GATK HaplotypeCaller pipeline is known

**Table 2. Resource usage across GermVarX pipeline stages. The complete pipeline finished in 77.8 hours of wall-clock time. Reported CPU and wall-clock hours are cumulative task sums across all samples in each stage, not actual elapsed runtime.**

| Pipeline Stage | No. Tasks | Total CPU Hours | Total Wall Clock Hours | Peak Memory (GB) |
|---|---|---|---|---|
| Preprocessing (FASTQ/BAM) | 720 | 585.8 | 226.3 | 20.7 |
| Variant Calling (DeepVariant) | 122 | 186.1 | 51.1 | 10.3 |
| Variant Calling (GATK) | 169 | 159.1 | 93.1 | 10.6 |
| Quality Assessment (FASTQC) | 240 | 19.8 | 19.7 | 0.4 |
| Coverage Analysis (mosdepth) | 120 | 3.3 | 3.7 | 1.9 |
| Annotation (VEP) | 1 | 1.5 | 0.4 | 3.3 |
| Variant Recalibration (GATK VQSR) | 1 | 0.2 | 0.1 | 4.9 |
| QC Variants (Bcftools) | 1 | 0.1 | 0.1 | 0.5 |
| Consensus Building | 1 | 0.0 | 0.0 | 0.0 |
| Reporting (MultiQC) | 1 | 0.0 | 0.0 | 0.7 |
| QC Samples (Plink) | 4 | 0.0 | 0.0 | 0.1 |

**Table 3. Comparison of GermVarX with existing WES workflows.**

| Feature | GermVarX | Sarek | MagicPipeline | star_protocols_wes |
|---|---|---|---|---|
| Primary Design Focus | Automated WES cohort workflow; germline variant discovery | WES/WGS per-sample workflow; germline & somatic variant discovery | WES cohort protocol; germline variant discovery | WES cohort protocol; germline variant discovery |
| Workflow Technology | Nextflow DSL2 | Nextflow DSL2 | Script-based; command-line dependent | Script-based; command-line dependent |
| Containerization | Yes (Docker) | Yes (Docker/Singularity) | No | Partial (Docker to run DeepVariant) |
| Variant Caller(s) | Dual caller: GATK HaplotypeCaller, DeepVariant | Multiple callers (GATK, Strelka, DeepVariant, Manta, etc.) | GATK | GATK HaplotypeCaller, FreeBayes, DeepVariant |
| Joint Genotyping | Yes | No (primarily per-sample) | Yes | Yes |
| Consensus Calling | Yes | Yes | No | Yes |
| Cohort-Level Variant Quality Control | Yes | No | Yes | No |
| Variant Annotation | Yes (VEP) | Yes (VEP, SnpEff, SnpSift) | Yes (VEP) | Yes (SnpEff, SnpSift) |
| Automated End-to-End Workflow | Yes (FASTQ→joint VCF→QC, annotation, MultiQC) | Yes (FASTQ→VCF→annotation, MultiQC) | Partial (fragmented scripts) | Partial (fragmented scripts) |
| Reproducibility | High | High | Low | Low |
| Scalability (medium→large cohorts) | High | High | Medium | Medium |
| Ease of Use (minimal config, automated orchestration) | High | High | Low | Low |
| Cloud & HPC Compatibility | Yes | Yes | Limited | Limited |

to perform best in large cohorts when paired with Variant Quality Score Recalibration (VQSR) [8,12]. As GermVarX scales, applying VQSR for GATK and leveraging larger cohorts should improve performance for both callers and the overall pipeline.

DV-GLN was particularly well suited for INDEL detection, showing superior precision compared to GATK alone. The consensus strategy, which retains only variants called by both pipelines, provided an intermediate balance—delivering very high precision with a modest reduction in recall. In the context of large cohort analyses, consensus calling offers a clear advantage: by prioritizing high-confidence variants consistently supported by independent algorithms, GermVarX reduces the risk of false positives that could confound downstream analyses such as association testing, polygenic risk score estimation, and rare variant burden testing. At the same time, GermVarX preserves single-caller callsets as optional outputs, enabling researchers to adjust the sensitivity–specificity trade-off according to their study objectives.

During development, we observed that GATK does not support parallelization for the `CombineGVCFs/GenomicsD-BImport` or `GenotypeGVCFs` steps. To overcome this limitation, GermVarX performs per-chromosome merging and joint genotyping, enabling efficient parallel execution and substantially reducing overall runtime. The computational benchmarks further highlight GermVarX's efficiency and scalability: processing 120 WES samples required less than 78 hours on a mid-range HPC cluster while maintaining a modest memory footprint (<21 GB). These results demonstrate that GermVarX is not only scalable to medium-sized studies but also well positioned for cohorts of thousands of samples. Its parallel execution model and containerized implementation ensure that GermVarX can be readily deployed across cloud or HPC environments.

Despite these strengths, several limitations remain. Although containerization with Docker ensures portability, extending support to Apptainer/Singularity would broaden usability on HPC clusters where Docker is restricted. Future optimizations could include support for storage-efficient formats such as CRAM, as well as automated cohort-level quality control steps (e.g., sample filtering, PCA, ancestry inference, population stratification adjustment).

In addition, we aim to benchmark GermVarX on publicly available large-scale cohorts to further stress-test scalability, as well as extend the evaluation to whole-genome sequencing (WGS) data to demonstrate the pipeline's flexibility beyond WES. We also plan to expand the workflow beyond germline SNP/INDEL detection by integrating structural variant and copy-number variation (CNV) calling modules, along with adding compatibility for long-read sequencing data. These extensions will enable more comprehensive variant discovery and further broaden GermVarX's applicability across diverse genomic analyses.

## Conclusion

GermVarX is an automated, Nextflow-based workflow for germline variant calling in Whole-Exome Sequencing cohort studies. By integrating state-of-the-art bioinformatics tools and employing a dual variant calling strategy with subsequent consensus, GermVarX reliably produces a single, high-quality multi-sample VCF file, a fundamental requirement for cohort-level genetic analyses. Its robust quality control framework, comprehensive variant annotation, and reproducible design make GermVarX a valuable tool for genetic researchers seeking to uncover the genetic basis of human health and disease.

## Supporting information

**S1 File. Step-by-step protocol, also available on protocols.io.**
(PDF)

**S1 Table. Accuracy measures for variant callers used in GermVarX across GIAB WES samples.**
(XLSX)

## Acknowledgments

We gratefully acknowledge the Centre for Informatics and Computing (VAST) for providing the high-performance computing (HPC) resources used to conduct the experiments in this study.

## Author contributions

**Conceptualization:** Thao Thi Phuong Nguyen.

**Data curation:** Ngoc Thi Minh Truong, Hanh Hong Ha.

**Formal analysis:** Ngoc Thi Minh Truong.

**Funding acquisition:** Thao Thi Phuong Nguyen, Dung Duc Nguyen, Hanh Hong Ha.

**Investigation:** Thao Thi Phuong Nguyen, Thuy Van Mai.

**Methodology:** Thao Thi Phuong Nguyen, Dung Duc Nguyen, Dung Khoi Nguyen.

**Project administration:** Thao Thi Phuong Nguyen, Hanh Hong Ha.

**Resources:** Hanh Hong Ha.

**Software:** Thao Thi Phuong Nguyen, Dung Duc Nguyen, Thuy Van Mai, Dung Khoi Nguyen, Tung Dang Nguyen.

**Supervision:** Dung Duc Nguyen, Hanh Hong Ha.

**Validation:** Thao Thi Phuong Nguyen, Thuy Van Mai, Dung Khoi Nguyen, Tung Dang Nguyen, Ngoc Thi Minh Truong, Trang Thi Ha Tran.

**Visualization:** Tung Dang Nguyen, Ngoc Thi Minh Truong.

**Writing – original draft:** Thao Thi Phuong Nguyen, Dung Khoi Nguyen, Ngoc Thi Minh Truong.

**Writing – review & editing:** Thao Thi Phuong Nguyen, Dung Duc Nguyen, Hanh Hong Ha, Trang Thi Ha Tran.

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
