## [Decision Letter · Decision Letter 0]

2 Dec 2025

PONE-D-25-50266GermVarX: A Robust Workflow for Joint Germline Variant Exploration in Whole-Exome Sequencing CohortsPLOS ONE

Dear Dr. Nguyen,

Thank you for submitting your manuscript to PLOS ONE. After careful consideration, we feel that it has merit but does not fully meet PLOS ONE’s publication criteria as it currently stands. Therefore, we invite you to submit a revised version of the manuscript that addresses the points raised during the review process.

We look forward to receiving your revised manuscript.

Kind regards,

Mainak Bardhan, MD

Academic Editor

PLOS ONE

3. We note you have not yet provided a protocols.io PDF version of your protocol and/or a protocols.io DOI. When you submit your revision, please provide a PDF version of your protocol as generated by protocols.io (the file will have the protocols.io logo in the upper right corner of the first page) as a Supporting Information file. The filename should be S1_file.pdf, and you should enter “S1 File” into the Description field. Any additional protocols should be numbered S2, S3, and so on. Please also follow the instructions for Supporting Information captions [https://journals.plos.org/plosone/s/supporting-information#loc-captions]. The title in the caption should read: “Step-by-step protocol, also available on protocols.io.”

Please assign your protocol a protocols.io DOI, if you have not already done so, and include the following line in the Materials and Methods section of your manuscript: “The protocol described in this peer-reviewed article is published on protocols.io (https://dx.doi.org/10.17504/protocols.io.[...]) and is included for printing purposes as S1 File.” You should also supply the DOI in the Protocols.io DOI field of the submission form when you submit your revision.

If you have not yet uploaded your protocol to protocols.io, you are invited to use the platform’s protocol entry service [https://www.protocols.io/we-enter-protocols] for doing so, at no charge. Through this service, the team at protocols.io will enter your protocol for you and format it in a way that takes advantage of the platform’s features. When submitting your protocol to the protocol entry service please include the customer code PLOS2022 in the Note field and indicate that your protocol is associated with a PLOS ONE Lab Protocol Submission. You should also include the title and manuscript number of your PLOS ONE submission.

Reviewers' comments:

Reviewer's Responses to Questions

**Comments to the Author**

1. Does the manuscript report a protocol which is of utility to the research community and adds value to the published literature?

Reviewer #1: Yes

Reviewer #2: Yes

Reviewer #3: Yes

2. Has the protocol been described in sufficient detail?

To answer this question, please click the link to protocols.io in the Materials and Methods section of the manuscript (if a link has been provided) or consult the step-by-step protocol in the Supporting Information files.

The step-by-step protocol should contain sufficient detail for another researcher to be able to reproduce all experiments and analyses.

Reviewer #1: Yes

Reviewer #2: Yes

Reviewer #3: Yes

3. Does the protocol describe a validated method?

Reviewer #1: Yes

Reviewer #2: Yes

Reviewer #3: Yes

4. If the manuscript contains new data, have the authors made this data fully available?

The PLOS Data policy requires authors to make all data underlying the findings described in their manuscript fully available without restriction, with rare exception (please refer to the Data Availability Statement in the manuscript PDF file). The data should be provided as part of the manuscript or its supporting information, or deposited to a public repository. For example, in addition to summary statistics, the data points behind means, medians and variance measures should be available. If there are restrictions on publicly sharing data—e.g. participant privacy or use of data from a third party—those must be specified.requires authors to make all data underlying the findings described in their manuscript fully available without restriction, with rare exception (please refer to the Data Availability Statement in the manuscript PDF file). The data should be provided as part of the manuscript or its supporting information, or deposited to a public repository. For example, in addition to summary statistics, the data points behind means, medians and variance measures should be available. If there are restrictions on publicly sharing data—e.g. participant privacy or use of data from a third party—those must be specified.requires authors to make all data underlying the findings described in their manuscript fully available without restriction, with rare exception (please refer to the Data Availability Statement in the manuscript PDF file). The data should be provided as part of the manuscript or its supporting information, or deposited to a public repository. For example, in addition to summary statistics, the data points behind means, medians and variance measures should be available. If there are restrictions on publicly sharing data—e.g. participant privacy or use of data from a third party—those must be specified.requires authors to make all data underlying the findings described in their manuscript fully available without restriction, with rare exception (please refer to the Data Availability Statement in the manuscript PDF file). The data should be provided as part of the manuscript or its supporting information, or deposited to a public repository. For example, in addition to summary statistics, the data points behind means, medians and variance measures should be available. If there are restrictions on publicly sharing data—e.g. participant privacy or use of data from a third party—those must be specified.

Reviewer #1: N/A

Reviewer #2: Yes

Reviewer #3: N/A

**5. Is the article presented in an intelligible fashion and written in standard English?**

Reviewer #1: Yes

Reviewer #2: Yes

Reviewer #3: Yes

6. Review Comments to the Author

Reviewer #1: My main concern is that the described pipeline appears to be primarily a combination of existing software tools, which raises questions about its novelty and added value. The authors are encouraged to provide a stronger justification for the development of this pipeline—specifically, to clarify what unique features, integrations, or improvements it offers compared to existing solutions, and to articulate how it will advance or benefit the research community.

The authors state in the introduction: “While a few pipelines such as MagicPipeline [5] and star protocols wes [6] support multi-sample processing, they are predominantly script-based and command-line dependent, lacking key features such as automated orchestration, modular parallel processing, and scalability across diverse computing infrastructures.” However, it remains unclear how the proposed pipeline substantively improves upon these existing tools. Furthermore, many researchers working with WES data already employ well-established, in-house pipelines tailored to their specific workflows. As such, the practical advantages and broader relevance of the proposed pipeline to the wider research community are not evident.

Below are several additional comments and questions regarding the manuscript.

1) Authors use a 3-sample WES benchmark dataset for validation, but it would be more informative if they could conduct validations on additional data sets.

2) On page 6, they state: "the highest 185 F1-score (0.9649), balancing exceptional precision (0.9994) with strong recall (0.9327). Please clearly define these 3 metrics.

3) Figure 2 needs a legend.

4) In the subsection "Variant Filtering and Quality Control" on page 5, are users able to modify the filtering thresholds? For example, instead of DP < 10, can the users change it to DP < 20? This was not very clear.

5) Can authors also explain if their pipeline can handle if the users input the intermediate files? For example, if the users already have gVCF files, can the pipeline bypass the previous steps and start with the gVCF files?

6) A major improvement to the current manuscript would be to demonstrate that the pipeline can handle two or more cohort-level WES datasets originating from different sources and generate a merged, harmonized dataset. This is a technically challenging task, and successfully implementing such functionality would significantly enhance the novelty and impact of the work. However, it is acknowledged that this capability may be beyond the current scope of the study.

Reviewer #2: The manuscript submitted by Nguyen et al. describes a workflow for variant calling of population WES data. Notably, the workflow includes a step for joint variant calling to produce a single cohort VCF. The manuscript is well written and very clear. However, I have some comments:

1) The authors state that existing workflows are designed for single-sample analysis, while their workflow is optimized for multi-sample analysis (L13). However, one of the existing workflows they cite, Sarek, states in their documentation that it is very easy to run multiple samples via a CSV samplesheet. It seems to me that the authors should not emphasize single vs multi sample design, but rather their implementation of joint calling.

2) Related to above, I believe that the authors could expand more on how their workflow is different than the existing workflows they listed, and why those differences are important. As it is written currently, the only real distinction the authors make is the difference in design for single or multi sample analysis, which is insufficient. The authors could consider making a table to list the differences to make it easier for readers (i.e. potential users) to determine if this workflow is right for their application.

3) The workflow Github repo looks well documented, for which I commend the authors. I would also like to see the inclusion of a test dataset that users can run through the workflow to make sure the workflow runs correctly on their system.

Reviewer #3: The narrative is well-written, clear, and linear. Each section transitions smoothly. The authors have combined dual-caller consensus and an automated joint calling workflow that addresses the gap between single-sample and multi-sample pipelines. However, I would like to give a few suggestions:

1. Add runtime comparisons with other public workflows. A qualitative comparison will strengthen the claim of scalability.

2. Future work could include the integration of structural variant and copy number variation (CNV) detection modules, along with extending the workflow to accommodate whole-genome sequencing (WGS) and long-read data for comprehensive variant discovery. This would enhance its applicability to diverse genomic analyses.

3. Figure captions could be expanded to interpret the figure rather than reviewing the section text.

7. PLOS authors have the option to publish the peer review history of their article (what does this mean?). If published, this will include your full peer review and any attached files.). If published, this will include your full peer review and any attached files.). If published, this will include your full peer review and any attached files.). If published, this will include your full peer review and any attached files.

...

Reviewer #1: No

Reviewer #2: No

Reviewer #3: No

---

## [Author Response · Author response to Decision Letter 1]

2 Feb 2026

Dear Dr. Mainak Bardhan and Reviewers,

We are pleased to resubmit the revised version of our manuscript, 'GermVarX: A Robust Workflow for Joint Germline Variant Exploration in Whole-Exome Sequencing Cohorts.'

We sincerely thank you for the constructive feedback, which has substantially improved the rigor and completeness of the manuscript.

In this revision, we have carefully addressed all comments by:

* Strengthening the Novelty and Motivation: We revised the Introduction and added a comparison table (Table 3) to clearly articulate the unique engineering and value proposition of GermVarX compared to existing workflows.

* Enhancing Scalability Documentation: We clarified the workflow's performance, including its support for GenomicsDBImport and the technical improvement of per-chromosome parallelization for joint genotyping.

* Improving Usability and Documentation: We updated the workflow to allow users to easily modify filtering thresholds via the configuration file and provided a link to a test dataset for verification. The S1_File is provided as generated by protocols.io (Private link for reviewers: https://www.protocols.io/private/E379FC52F62011F0BBA60A58A9FEAC02).

* Addressing Accuracy Metrics: We added clear definitions for Precision, Recall, and F1-score.

A detailed, point-by-point response to all comments is provided in the uploaded 'ResponsetoReviewers.docx' file.

Thank you very much for your time and consideration. As the outcome of this review is time-sensitive for our ongoing research, we would greatly appreciate your expedited consideration of this revised manuscript. We look forward to hearing from you.

Sincerely,

Thao Nguyen

---

## [Decision Letter · Decision Letter 1]

9 Mar 2026

GermVarX: A Robust Workflow for Joint Germline Variant Exploration in Whole-Exome Sequencing Cohorts

PONE-D-25-50266R1

Dear Dr. Thao Nguyen,

We’re pleased to inform you that your manuscript has been judged scientifically suitable for publication and will be formally accepted for publication once it meets all outstanding technical requirements.

Kind regards,

Mainak Bardhan, MD

Academic Editor

PLOS One

Additional Editor Comments (optional):

Reviewers' comments:

Reviewer's Responses to Questions

**Comments to the Author**

1. Does the manuscript report a protocol which is of utility to the research community and adds value to the published literature?

Reviewer #1: Yes

Reviewer #3: Yes

2. Has the protocol been described in sufficient detail?

To answer this question, please click the link to protocols.io in the Materials and Methods section of the manuscript (if a link has been provided) or consult the step-by-step protocol in the Supporting Information files.

The step-by-step protocol should contain sufficient detail for another researcher to be able to reproduce all experiments and analyses.

Reviewer #1: Yes

Reviewer #3: Yes

3. Does the protocol describe a validated method?

Reviewer #1: Yes

Reviewer #3: Yes

4. If the manuscript contains new data, have the authors made this data fully available?

The PLOS Data policy requires authors to make all data underlying the findings described in their manuscript fully available without restriction, with rare exception (please refer to the Data Availability Statement in the manuscript PDF file). The data should be provided as part of the manuscript or its supporting information, or deposited to a public repository. For example, in addition to summary statistics, the data points behind means, medians and variance measures should be available. If there are restrictions on publicly sharing data—e.g. participant privacy or use of data from a third party—those must be specified.requires authors to make all data underlying the findings described in their manuscript fully available without restriction, with rare exception (please refer to the Data Availability Statement in the manuscript PDF file). The data should be provided as part of the manuscript or its supporting information, or deposited to a public repository. For example, in addition to summary statistics, the data points behind means, medians and variance measures should be available. If there are restrictions on publicly sharing data—e.g. participant privacy or use of data from a third party—those must be specified.requires authors to make all data underlying the findings described in their manuscript fully available without restriction, with rare exception (please refer to the Data Availability Statement in the manuscript PDF file). The data should be provided as part of the manuscript or its supporting information, or deposited to a public repository. For example, in addition to summary statistics, the data points behind means, medians and variance measures should be available. If there are restrictions on publicly sharing data—e.g. participant privacy or use of data from a third party—those must be specified.requires authors to make all data underlying the findings described in their manuscript fully available without restriction, with rare exception (please refer to the Data Availability Statement in the manuscript PDF file). The data should be provided as part of the manuscript or its supporting information, or deposited to a public repository. For example, in addition to summary statistics, the data points behind means, medians and variance measures should be available. If there are restrictions on publicly sharing data—e.g. participant privacy or use of data from a third party—those must be specified.

Reviewer #1: N/A

Reviewer #3: N/A

**5. Is the article presented in an intelligible fashion and written in standard English?**

Reviewer #1: Yes

Reviewer #3: Yes

6. Review Comments to the Author

Reviewer #1: The authors have addressed the previous comments adequately, and the current reviewer does not have any additional comments.

Reviewer #3: The authors have addressed most of the comments, including those related to accuracy and computational performance.

They have also included an additional dataset of 120 samples of WES cohort to demonstrate the workflow’s reliability and accuracy.

The authors have also addressed the key differences between GermVarX and existing WES workflows.

Overall, the manuscript is significantly improved.

7. PLOS authors have the option to publish the peer review history of their article (what does this mean?). If published, this will include your full peer review and any attached files.). If published, this will include your full peer review and any attached files.). If published, this will include your full peer review and any attached files.). If published, this will include your full peer review and any attached files.

...

Reviewer #1: **Yes:** Harold BaeHarold BaeHarold BaeHarold Bae

Reviewer #3: No

---

## [Editor Report · Acceptance letter]

PONE-D-25-50266R1

PLOS One

Dear Dr. Nguyen,

I'm pleased to inform you that your manuscript has been deemed suitable for publication in PLOS One. Congratulations! Your manuscript is now being handed over to our production team.

Kind regards,

on behalf of

Dr. Mainak Bardhan

Academic Editor

PLOS One